# LongCodeBench: Evaluating Coding LLMs at 1M Context Windows

**Stefano Rando**[*]**& Yuta Kyuragi**
Panasonic AI Research
{stefano.rando, yuta.kyuragi}@us.panasonic.com

**Luca Romani**[*] **& Fabio Galasso**
Sapienza University of Rome
luca.romani@uniroma1.it
galasso@di.uniroma1.it

**Alessio Sampieri**[*] **& Luca Franco**
ItalAI
{alessio.sampieri, luca.franco}@italailabs.com

**John Yang & Tatsunori Hashimoto**
Stanford University
{johnby, thashim}@stanford.edu

## Abstract

Context lengths for models have grown rapidly, from thousands to millions of tokens in just a few years. The extreme context sizes of modern long-context models have made it difficult to construct realistic long-context benchmarks – not only due to the cost of collecting million-context tasks but also in identifying realistic scenarios that require significant contexts. We identify code comprehension and repair as a natural testbed and challenge task for long-context models and introduce **LongCodeBench** (**LCB**), a benchmark to test LLM coding abilities in long-context scenarios. Our benchmark tests both the comprehension and repair capabilities of LCLMs in realistic and important settings by drawing from real-world GitHub issues and constructing QA (**LongCodeQA**) and bug fixing (**LongSWE-Bench**) tasks. We carefully stratify the complexity of our benchmark, enabling us to evaluate models across different scales – ranging from Qwen2.5 14B Instruct to Google's flagship Gemini model. We find that long-context remains a weakness for all models, with performance drops such as from 29% to 3% for Claude 3.5 Sonnet, or from 70.2% to 40% for Qwen2.5.

## 1 Introduction

Long-context modeling has arisen as an active research area, motivated by the potential of models capable of processing extended inputs in real-world applications, such as entire code repositories or large document collections. This trend is reflected in the emergence of Large Context Language Models (LCLMs) (Anthropic, 2024; OpenAI, 2024; GeminiTeamGoogle, 2024; JambaTeam, 2024), with industry-driven models now supporting context windows up to millions of tokens. In parallel, experimental architectures continue to explore novel approaches for long-range modeling (Gu et al., 2022; Gu & Dao, 2024). As illustrated in Figure 1, model context lengths have grown superexponentially in recent years, making it important for us to understand whether and how such context sizes are effective.

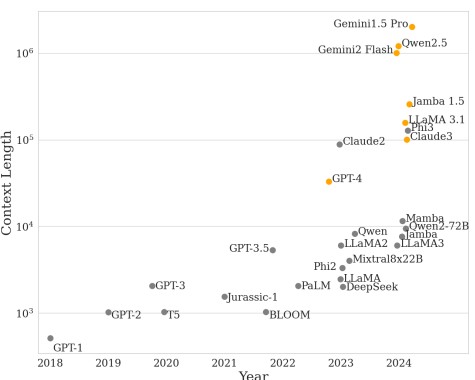

Figure 1: Increasing trend of LCLM context lengths over time. Models tested on LCB are highlighted in orange.

Evaluation of million-token LCLMs has been challenging, with many works relying on synthetic tests (Tay et al., 2021; Kamradt, 2023)

---

[*]Equal contribution

| | Tasks | | | Principles | | |
|---|---|---|---|---|---|---|
| | Comprehension | Repair | Coding | Synthetic | Max. Context | Granular Eval. |
| LRA | ✓ | ✗ | ✗ | Yes | 16K | ✗ |
| NIAH | ✓ | ✗ | ✗ | Partial | 200K | ✗ |
| RULER | ✓ | ✗ | ✗ | Yes | 200K | ✗ |
| HELMET | ✓ | ✓ | ✗ | Partial | 128K | ✓ |
| ∞BENCH | ✓ | ✗ | ✓ | Yes | 256K | ✗ |
| SWE-Bench | ✗ | ✓ | ✓ | No | 50K | ✗ |
| LongCodeBench | ✓ | ✓ | ✓ | No | 1M | ✓ |

Table 1: Comparison of long-context benchmarks across task coverage (left) and core principles (right). *Repair* tasks support **Scalability**. *Generation* tasks, *coding*, and *non-synthetic data* reflect **Realism**. *Maximum context length* and *granular evaluation* assess **Long-context**. See the discussions in Sections 1 and 3 for details.

and a few long-context reading comprehension style tasks (Wu et al., 2025; Wang et al., 2024; Hsieh et al., 2024). However, these tasks do not fully capture the transformative potential of LCLMs envisioned by their creators, where LCLMs are imagined to learn a foreign language from a single book or find and fix bugs after ingesting an entire codebase (GeminiTeam-Google, 2024).

In this work, we focus on the last of these scenarios, aiming to build a benchmark that allows us to track progress in LMs that understand and fix codebases in challenging real-world scenarios. This benchmark setting is underexplored, with most current coding benchmarks (Jimenez et al., 2024; Bogomolov et al., 2024) evaluating shorter context lengths (up to 64K–200K tokens) or focus on isolated tasks such as commit message generation. Our work expands upon these existing benchmarks by testing the largest existing contexts and grounding the tasks and scenarios in real-world, economically valuable software engineering tasks.

We propose **LongCodeBench** (**LCB**), a benchmark that evaluates coding LCLMs on both code comprehension and repair across a range of context sizes, ranging from tens of thousands of tokens up to one million. To test code comprehension, we construct **LongCodeQA**, which tests comprehension by posing questions derived from real GitHub issue discussions. We complement this task with **LongSWE-Bench**, which assesses repair through debugging tasks that require models to generate patches for bugs of actual software. As illustrated in Table 1 (*left*), our benchmark is grounded on real-world tasks, while providing coverage over tasks and context lengths. The dataset comprises 1043 instances that have been collected from 108 repositories, and have been curated with human supervision to achieve high-quality standards.

As illustrated in Table 1 (*right*), we argue for three principles that guide the creation of LCB:

- **Scalability.** Despite recent advancements, most LCLMs still struggle to achieve reliable performance in long-context settings—especially in coding, where evaluation is often binary: a generated code either works or fails. To provide more informative and graded feedback, LCB includes the QA task on which weaker LCLMs can achieve nonzero performance to serve as a "gradient" for the LCLM improvement. This enables LCB to serve as a useful benchmark for both frontier models and smaller research models.

- **Realism.** Coding is one of the most practically grounded applications of LCLMs, where models are expected to support real-world developer workflows. To reflect this, all data in LCB are drawn from public GitHub repositories, avoiding synthetic data or artificially simplified tasks. In particular, both LongSWE-Bench and LongCodeQA are derived from GitHub issues that software engineers raised for real-world problems. This ensures that LCB provides realistic evaluation settings aligned with how LCLMs may be used in the future.

- **Long-context.** Modern software repositories often span thousands to millions of tokens, requiring models to process entire codebases for many tasks. LCB introduces

input contexts up to one million tokens—the longest in existing coding benchmarks. Since model performance is influenced by both architecture and context length, we also provide evaluations at multiple scales (32K, 64K, 128K, 256K, 512K, and 1M). This design enables fine-grained analysis of long-context behavior, helping to disentangle context scaling effects from model capabilities, while also ensuring accessibility for resource-constrained evaluations at smaller context sizes.

We evaluate several recent LCLMs on LCB to quantify their performance in fundamental software engineering practices (e.g., bug fixing and code comprehension). Our analysis reveals important trends: performance degradation with increasing context lengths, inability to infer performance at longer contexts from shorter ones, and correlation between length sensitivity and task complexity. Ultimately, we show that long-context remains a challange even for the best models: The accuracy of Claude 3.5 Sonnet on LongSWE-Bench drops from 29% to 3% when increasing context length from 32K to 256K. The analysis informs good practices on future benchmarking of LCLMs, while indicating LCB as a suitable benchmark that enables those practices.

The LCB dataset is available publicly at https://huggingface.co/datasets/Steefano/LCB.

## 2 Related work

### 2.1 Benchmarks for LCLMs

Recent benchmarks have increasingly extended the context lengths, identifying the necessity in real-world applications to process entire documents, large code repositories, or multi-file inputs. RULER (Hsieh et al., 2024) and ∞BENCH (Zhang et al., 2024) extend context windows beyond 128K tokens to push the limits of LLMs in retrieving and understanding long contextual information. Similarly, Needle-in-a-Haystack (NIAH) (Kamradt, 2023) tests for long-context recall by querying models to detect the position of a text fragment ("needle") inserted within a long document ("haystack"). Long Range Arena (Tay et al., 2021), one of the earliest datasets targeting long-range dependencies, addresses this challenge, but it remains limited to synthetic tasks that do not reflect the LCLMs' real-world effectiveness. Recent works have focused on grounding long-context evaluation in real-world data to overcome the limitations of synthetic benchmarks. HELMET (Yen et al., 2025) improves task realism by collecting data samples from practical domains, such as legal or scientific documents. While this shift enhances applicable relevance, some proposed tasks remain artificial, including synthetic recall and passage reranking.

Another key trend is the diversification of tasks to assess a broader range of model capabilities. Benchmarks like HELMET, RULER, and NIAH evaluate skills such as complex reasoning, in-depth summarization, and robust long-range dependency understanding. By testing models on multiple tasks, these benchmarks offer a more complete evaluation of model capabilities in different contexts. Our LCB complements existing benchmarks while addressing gaps in long-context evaluation by (i) targeting long-context evaluation, aligning with the broader push toward handling extended sequences of text; (ii) focusing on real-world tasks, with an emphasis on practical software engineering scenarios rather than synthetic setups; and (iii) introducing novel coding tasks, such as long-context bug localization and code comprehension via question-answering, which are underrepresented in existing benchmarks. In this way, LCB extends the long-context evaluation landscape into the software engineering domain — a critical area for real-world applications yet underexplored in existing benchmarks.

### 2.2 Coding benchmarks for LCLMs

Coding has become one of the most widely adopted applications for LLMs, as demonstrated by the growing number of specialized benchmarks. A seminal work is HumanEval (Chen et al., 2021), which provides hand-written coding problems where models generate code from natural language docstrings and evaluate correctness with functional tests. Building on HumanEval, several works (Cassano et al., 2023; Orlanski et al., 2023) have extended their scope by expanding the language coverage and the evaluation tasks, as well as increasing

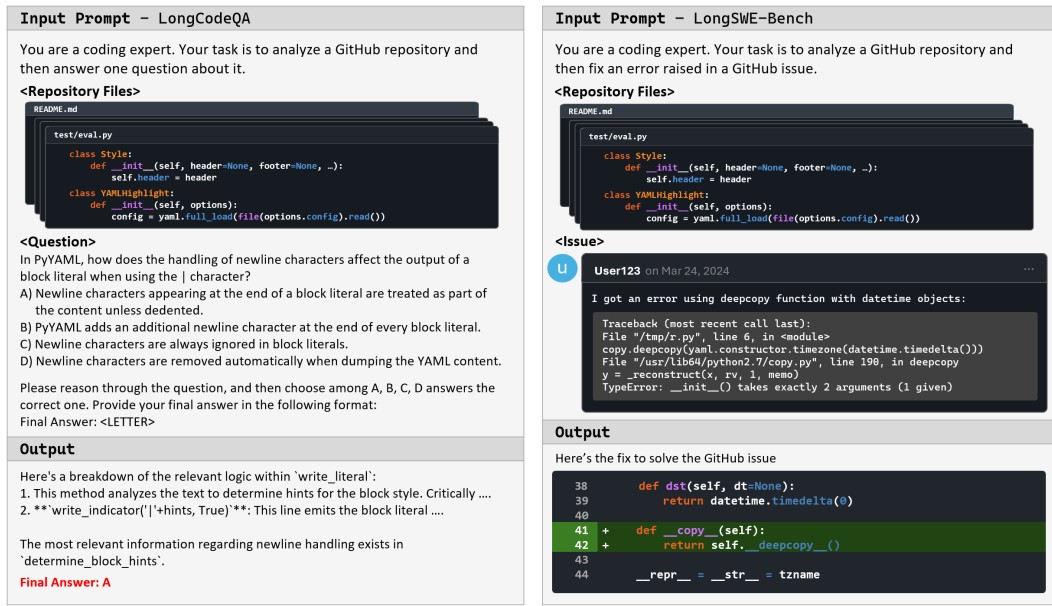

Figure 2: Input prompt structure and output format for the two LCB tasks. LongCodeQA (left) answer a multiple-choice question considering the full repository and the question derived from GitHub issues. LongSWE-Bench (right) generate a bug-fixing patch considering a subset of the codebase files and the GitHub issue.

the sample size and diversity. Similarly, RepoBench (Liu et al., 2024) draws data from public GitHub repositories and departs from execution-based metrics, a significant shift that underlines the need for alternative evaluation criteria.

SWE-Bench (Jimenez et al., 2024) proposes a range of debugging tasks by testing models on real-world problems extracted from public GitHub issues with minimal processing overhead, thereby closely approximating actual debugging scenarios. Inspired by SWE-Bench realism, LongCodeBench proposes similar debugging tasks while also introducing a novel challenge for code comprehension, which is missing in the mentioned benchmarks. Moreover, LCB pushes the context length well beyond the 50K token limit of SWE-Bench, providing samples of up to a million tokens. This enables a more challenging and comprehensive evaluation framework for coding applications.

# 3    LongCodeBench

In this section, we introduce an overview of the proposed benchmark. Section 3.1 outlines the general design principles, while Sections 3.2 and 3.3 detail the structure of the LongCodeQA and LongSWE-Bench tasks, respectively. For each task, we report its definition and evaluation, prompt format, data collection process, and data statistics.

## 3.1    Code comprehension and repair

Code comprehension and repair test complementary capabilities of LCLMs. Testing on these two tasks, LCB assesses the LCLM performances in extended contexts under different perspectives:

- **Task-context interaction.** Both tasks in LCB operate under the same long-context regime, but they place fundamentally different demands on how models interact with the context. In the comprehension task (LongCodeQA), models must identify relevant information to answer a specific question, evaluating both their implicit comprehension of the repository-level context and their explicit reasoning skills for

|  |  | 32K | 64K | 128K | 256K | 512K | 1M |
|---|---|---|---|---|---|---|---|
| LongCodeQA | # Instances | 113 | 76 | 92 | 65 | 47 | 50 |
|  | # Repositories | 27 | 16 | 20 | 15 | 11 | 9 |
|  | # Files/Repo | 26 | 56 | 88 | 93 | 209 | 202 |
|  | # Tokens/File | 690 | 862 | 1047 | 1840 | 1782 | 3419 |
| LongSWE-Bench | # Instances | 100 | 100 | 100 | 100 | 100 | 100 |
|  | # Repositories | 12 | 11 | 11 | 10 | 10 | 7 |
|  | # Files/Repo | 8 | 17 | 27 | 62 | 135 | 203 |
|  | # Tokens/File | 5200 | 3668 | 4646 | 4219 | 3211 | 4372 |

Table 2: Dataset statistics for the two LCB tasks across different context-length brackets. For each bracket (32K to 1M), we report: the number of instances and repositories considered, the average number of files per repository and the average number of tokens per file.

targeted information extraction. In contrast, the generation task (LongSWE-Bench) expects the model to produce a valid code patch consistent with the existing project structure. This dichotomy allows LCB to evaluate both content understanding and active integration within long-context sequences.

- **Attention distribution.** LCB's task design provides an opportunity to examine how models distribute their attention when processing long-context inputs. LongCodeQA questions require inspecting the repository for definitions, tracing function objectives, and usage. LongSWE-Bench, on the other hand, emphasizes depth: the model must anchor its generation in the local project logic, synthesizing precise, executable edits. These tasks stress different computational bottlenecks—information retrieval versus generation precision—offering complementary insights into LCLMs behavior scale.

For long-context benchmarks, dataset size must be carefully balanced against the tradeoff between inference cost and statistical reliability. As the input scales up to a million tokens, resource requirements scale up quickly, risking inaccessibility if the dataset includes too many instances, while a sufficient and representative number of samples is needed for robust evaluation. In LCB, we collect a sufficient number of samples to ensure robust estimation while containing the evaluation costs within practical limits. We report the total cost and time of inference for the models tested on LCB in Table 6.

## 3.2 LongCodeQA

Here we detail LongCodeQA, including the task formulation, the prompt structure, and data information (i.e., collection, validation, and statistics).

**Task formulation** To test the models' code comprehension ability in a long-context scenario, we propose LongCodeQA, which is a multiple-choice question task. Each question is derived from public GitHub Python repositories, grounding the benchmarking in real-world software engineering scenarios.

The evaluation metric considered is accuracy, which is measured as the percentage of correct responses over the total number of questions.

**Prompt structure** Figure 2 (*left*) shows an example of the input prompt structure and the response format. The prompt contains:

- **Instructions.** Description of the QA task.
- **Repository files.** The full repository to which the question is related, with files sorted alphabetically by their name.
- **Question.** Multiple-choice question.

To ensure fairness and avoid positional biases in model evaluation, the correct answer is randomly assigned to one of the four multiple-choice options, with uniform shuffling

applied during generation. Moreover, the task is presented in a zero-shot format, with no additional instructions or in-context examples provided to the prompt. Complete prompt examples are reported in appendix B.

**Data collection**    Collecting data for LongCodeQA involves different steps: selecting the relevant repositories, filtering the issues, and converting issues to multiple-choice questions.

As a first step, to ground the LongCodeQA task in real-world scenarios, we collect questions derived from closed GitHub issues via GitHub REST APIs. Repositories are selected based on their total token length to ensure a consistent distribution of samples across different context-length brackets. To have a significant number of samples, we prioritize repositories with a high number of publicly available issues.

After selecting repositories based on the mentioned criteria, the second step involves filtering for relevant issues. To achieve this, we leverage an LLM (i.e., GPT-4) prompting it to filter out cases involving bug fixing, new feature addition, installation problems, project roadmaps, or development practices. For the selected issues, the prompt instructs the LLM to convert them into a multiple-choice question, with the use of structured outputs (Willard & Louf, 2023) for robust formatting. However, we notice that 64% of the questions generated can be answered using general coding knowledge alone, without referencing the repository content. As a final safeguard against data contamination (Balloccu et al., 2024), we prompt GPT-4 to answer each question in the dataset without access to context, and we filter out the questions on which the model succeeded using internal knowledge alone. After all iterations, the percentage of initial issues that are converted into appropriate questions is 3.99%. This small percentage reinforces our first point about prioritization of repositories with a sufficient number of issues.

As a final step, we generate the specific question from the issue, the correct answer, and the other three wrong answers. The LLM performs an additional step in extracting questions and answers.

**Data validation**    We design the data generation pipeline for LongCodeQA to satisfy two key criteria: *reliability* and *fairness*. To assess *reliability*, we perform manual verification on a stratified random subset of 9 questions for each context length bracket. We verify that for 52 of the 54 selected questions—96.3%—in-depth research about the repository and its internal codebase is required to answer successfully. Regarding *fairness*, a central design decision in our pipeline is to generate all questions using only the issue "Discussion", without referencing the full codebase. This ensures that the benchmark is unbiased and does not benefit from any implicit access to repository content during construction, preserving its role as a fair evaluation tool for data construction.

**Dataset statistics**    Table 2 (*top*) reports additional statistics to characterize the structure and variability of the LongCodeQA dataset. The task comprises 443 question-answer instances drawn from 98 public GitHub repositories. Shorter repositories are more common on GitHub, while larger ones are comparatively rare. To ensure balanced evaluation across context lengths, we sample a representative number of QA instances for each bracket. The first and second rows of Table 2 report, respectively, the final number of instances and repositories contributing to each bracket.

To better understand what drives context size, we analyze the repository structure. The overall context length of each repository depends on both the average number of files per repository and the average number of tokens per file, reported in the third and fourth rows of Table 2, respectively. These two factors reflect complementary sources of complexity: a higher number of files increases inter-file dependencies, while longer individual files introduce more intricate intra-file dependencies. We perform an analysis on this aspect in Section 4, as illustrated in Figure 3.

### 3.3   LongSWE-Bench

As for LongCodeQA, this section describes the LongSWE-Bench task, describing the task formulation, the prompt structure, and data information (i.e., collection, validation, and statistics).

**Task formulation**    LongSWE-Bench tests a model's code repair capabilities in a realistic setting, specifically targeting the task of bug fixing. Each instance is derived from real-world GitHub issues in public Python repositories, requiring the model to fix a codebase error by considering a long-context input consisting of multiple repository files.

We employ an execution-based metric for evaluation, validating the generated code by running unit tests associated with the input issue. A model response is considered valid if all unit tests pass, while any failed test or compilation error counts as a failure. We report performance as the percentage of solved issues, effectively reflecting accuracy in the real-world debugging context.

**Prompt structure**    Figure 2 (*right*) presents an example of the input prompt structure and the output response format. The prompt contains:

- **Instructions.** Description of the task and its requirements;
- **Repository files.** Structured as two sets of files:
  - **Ground-truth files** from the pull request that fixed the issue and additional;
  - **Random files** sampled from the repository;
- **Issue information.** Including the title and body of the issue;
- **Response format.** Illustrating how the model should structure its answer.

We regulate the prompt length by selecting a subset of files (ground-truth and random) aligning with the desired target context length. No additional instructions or in-context examples are included, making the task zero-shot. We report additional prompt details in Appendix B.

**Data collection**    We assemble LongSWE-Bench through three iterative data refining steps. In the first step, we select public Python GitHub repositories, that meet quality and size criteria. For quality, we prioritize repositories with a large number of reported issues and an active development cycle, as these offer a richer set of real-world debugging cases. For size, we select repositories of sufficient length, ensuring that designated issues come from projects with considerable codebase.

In the second step, we filter for issues that have been resolved through a pull request. However, not all such issues are suitable: many lack explicit test cases or involve refactoring rather than a clear bug fix. Then we restrict the selection to issues for which unit tests are available, guaranteeing a more reliable performance metric.

As a third step, we manually discard ill-posed samples. Issues have to include a clear description of the problem and a well-defined solution within the provided information, excluding feature requests or ambiguous bug reports.

**Data validation**    We design the data collection pipeline to ensure two criteria: *reliability* and *reproducibility*. To ensure the benchmark *reliability*, we evaluate all the instances in the benchmark manually, ensuring that they are well-posed and unambiguous. We do not include issues involving the addition of new features or design choices, as those correspond to more open-ended problems that are challenging to evaluate. To further verify data reliability, we provide a detailed analysis in Appendix A, which guides the design of the issue filtering pipeline used during data collection. To ensure *reproducibility*, we create a dedicated execution environment to replicate each issue, guaranteeing that tests run on the same version of the codebase in which the issue was originally raised. This is accomplished by creating a dedicated Docker image per instance, where we install the correct dependencies and verify the bug's presence before applying the patch. This approach ensures a reproducible and reliable execution of the unit tests.

**Dataset statistics**    Table 2 (*bottom*) provides statistics that characterize the structure and variability of the LongSWE-Bench dataset. The task comprises 600 bug-fixing instances evenly distributed across six context-length brackets (32K to 1M), with 100 instances per bracket. These instances are sampled from 61 public Python repositories. Moreover, we analyze the repository composition, reporting both the number of files and the average token length per file.

| | LongCodeQA | | | | | | LongSWE-Bench | | | | | |
|---|---|---|---|---|---|---|---|---|---|---|---|---|
| | 32K | 64K | 128K | 256K | 512K | 1M | 32K | 64K | 128K | 256K | 512K | 1M |
| Qwen2.5 - 14B Instruct | 61.9 | 65.8 | 68.5 | 63.1 | 70.2 | 40.0 | 0 | 0 | 0 | 0 | 0 | 0 |
| Jamba 1.5 - 400B Large | 69.0 | 69.7 | 72.8 | 54.2 | - | - | 3 | 1 | 1 | 0 | - | - |
| Llama 3.1 - 405B Instruct | 69.9 | 72.4 | 67.4 | - | - | - | 0 | 1 | 0 | - | - | - |
| Llama 4 Scout | 66.4 | 73.7 | 70.7 | 63.1 | 78.7 | 76.0 | 0 | 0 | 0 | 0 | 0 | 0 |
| GPT-4o | 65.5 | 76.3 | 74.3 | - | - | - | 11 | 6 | 5 | - | - | - |
| GPT-4.1 | 72.6 | 73.7 | 78.3 | 72.3 | 78.7 | 80.0 | 1 | 1 | 1 | 1 | 1 | 2 |
| Gemini 2 Flash | 66.4 | 68.4 | 65.2 | 63.1 | 70.2 | 65.5 | 10 | 6 | 7 | 3 | 2 | 2 |
| Gemini 1.5 Pro | 67.3 | 63.2 | 72.8 | 64.6 | 72.3 | 66.0 | 1 | 6 | 2 | 3 | 4 | 5 |
| Gemini 2.5 Pro | 75.2 | 71.1 | 71.7 | 67.7 | 68.1 | 69.8 | 23 | 25 | 22 | 24 | 12 | 7 |
| Claude 3.5 Sonnet | 65.5 | 69.7 | 71.7 | 66.6 | - | - | 29 | 19 | 15 | 3 | - | - |
| Claude 3.5 Sonnet + RAG | 25.55 | 31.18 | 24.83 | 25.95 | 17.19 | 12.77 | 23 | 23 | 22 | 21 | 16 | 22 |

Table 3: Performance of different models on LongCodeQA and LongSWE-Bench across increasing context lengths (32K to 1M tokens). LongCodeQA is evaluated by answer accuracy, while LongSWE-Bench is evaluated by the percentage of solved issues. The table's top section lists open-source models, while closed-source models appear below, and finally a RAG based model is reported in the last row.

# 4 Analysis

This section presents the evaluation results for a range of open- and closed-source models detailed in Section 4.1. The results for LongSWE-Bench and LongCodeQA tasks are discussed relatively in Sections 4.2 and 4.3. Section 4.4 reports additional observations and insights based on the results.

## 4.1 Selection of LCLMs

We evaluate diverse open- and closed-source models supporting context windows up to 1M tokens. Our selection prioritizes overall performance, maximum context length, and architectural diversity. Among closed-source models, we include GPT-4o (Transformer, 128K) (OpenAI, 2024), GPT-4.1 (Transformer, 1M), Claude 3.5 Sonnet (Transformer, 256K) (Anthropic, 2024), Gemini 2 Flash, 1.5 Pro, and 2.5 Pro (Transformer and MoE, 1M) (GeminiTeam-Google, 2024). In the open-source category, we consider Qwen2.5-14B-Instruct (Transformer, 1M) (QwenTeam, 2024), Jamba 1.5 400B Large (hybrid Transformer-Mamba with State Space Model components and MoE, 256K) (JambaTeam, 2024), and Llama 3.1 405B Instruct and 4 Scout (Transformer, 128K-1M) (LlamaTeam, 2024). This selection underscores current performance and architectural innovations in long-context language models.

In this work, we deliberately focus on long-context evaluation rather than RAG optimization. We view these as distinct approaches to handling large documents and LongCodeBench aims to test pure long-context capabilities. For reference, we report results for Claude 3.5 Sonnet with RAG. We remark that performance of RAG architectures depends on prompt structure and retrieval techniques and therefore the scores reported here are a narrow view of RAG capabilities. We report each model's detailed inference time and cost in Appendix C.

## 4.2 Performance on LongCodeQA

We report LongCodeQA results across the different context length brackets in Table 3 (*left section*). Most models demonstrate competitive performance at short to mid-range brackets (32K–128K), with accuracy peaking between 64K and 128K for nearly all models. For example, Llama 3.1 achieves the maximum accuracy at 64K with 72.4%, and GPT-4o reaches 76.3%, the highest result in the short-mid range. Claude 3.5 Sonnet and Jamba maintain the most solid performance, all above 69% across the lower context windows. Qwen2.5 steadily improves from 61.9% at 32K to a peak of 70.2% at 512K, showing stability in longer contexts—though this trend drops drastically at 1M, where accuracy reduces at 40.0%. Similarly, Jamba has good performance up to 128K but degrades significantly at

256K, with accuracy dropping from 72.8% to 54.2%. GPT-4.1 has the best results on the task, especially showing an improvement over its predecessor, GPT-4o, and consistency across context lengths. Llama 4 also improves over Llama 3.1, with an higher context window and better performance at 64K and 128K. Gemini 2 Flash, Gemini 1.5 Pro, and Gemini 2.5 Pro show remarkably stable behavior across all brackets up to 1M, ranging between 63–72%, preserving their accuracy in extremely long contexts. We observe that RAG negatively impacts the performance of Claude 3.5 Sonnet, achieving a score comparable to random guessing. This is an interesting phenomenon that shows the sensitivity of RAG algorithms to different tasks.

Overall, LongCodeQA shows that model performance at increasing context lengths is neither monotonic nor uniform, with several models peaking early or degrading drastically. This supports the need for long and granular evaluations like those in LCB, which expose weaknesses in long-context handling and emphasize the gap between theoretical context support and actual robustness.

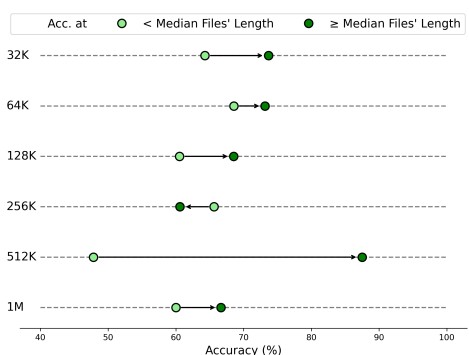

Figure 3: Accuracy of Gemini 2 Flash on LongCodeQA split into two subsets: samples whose prompts have an average # Tokens/File below or above median.

### 4.3 Performance on LongSWE-Bench

We report LongSWE-Bench results across context brackets in Table 3 (*right section*), measuring the number of successfully resolved issues per model per bracket. As expected from the difficulty of the task—real-world bug fixing requires precise code patch—the results are significantly lower compared to LongCodeQA. Despite the overall difficulty, some trends emerge. Claude 3.5 Sonnet shows the strongest performance at shorter context lengths, solving 29% of issues at 32K. It maintains relatively high success at 64K and 128K but drops at 3% sharply at 256K. Both Gemini 2 Flash and GPT-4o peak at 32K with 11% and 10% relatively and show a gradual drop across longer contexts. On the other hand, Gemini 1.5 Pro demonstrates a consistently low performance, solving between 1% and 6% across the full range; Gemini 2.5 Pro approaches Claude 3.5 at the 32K window with 23% and maintaining a consistent rate of success until 256K, decreasing to 7% at 1M. Open-source models perform significantly worse. Jamba solves up to 3% issues at 32K but drops to 0% by 256K. Qwen2.5, Llama 3.1, Llama 4, and GPT-4.1 fail to solve any issue across all tested brackets. Contrary to the LongCodeQA task, RAG helps Claude 3.5 Sonnet achieve positive scores on LongSWE-Bench, stressing again the importance of RAG strategy dependent on the deployment task. These results reflect both the increased challenge of generative code tasks and the current gap between open- and closed-source models in long-context code repair.

LongSWE-Bench exposes the limitations of current LCLMs in long-context settings. While some models perform reasonably well at shorter context lengths, their effectiveness declines significantly as the context increases — an effect amplified by the binary nature of the task, where even small errors lead to complete failure.

### 4.4 Discussion

**Specialized knowledge** Table 3 reveals insightful trends when comparing earlier LLM versions—Llama 3.1 405B Instruct, GPT-4o, and Gemini 1.5 Pro—with their more recent counterparts: Llama 4 Scout, GPT-4.1, and Gemini 2.5 Pro. While Llama 4 Scout and GPT-4.1 show improved performance on LongCodeQA, they underperform on LongSWE-Bench; e.g. GPT-4.1 accuracy drops from a peak of 11% to one of 2%. Llama 4 Scout fails to address the task at every context length regime, similarly to its predecessor. Conversely, Gemini 2.5 Pro displays a substantial gain on LongSWE-Bench (7–25% compared to 6–1% for its predecessor), but sees no comparable improvement on LongCodeQA. These trajectories

suggest a possible shift in model development toward specialized competence. Rather than exhibiting consistent gains across tasks, newer models appear increasingly optimized for particular domains or problem types—raising the possibility that improvements in long-context reasoning come at the expense of general-purpose adaptability.

**File length vs. performance** Table 2 shows that the benchmark's brackets vary in the total context length and the average tokens per file. Then, we analyze the correlation between file length and performance (Figure 3). For each bracket in LongCodeQA, we calculate the median average file length per sample and split the bracket into two subsets—one below and one above the median. We then evaluate Gemini 2 Flash's accuracy on both subsets, observing that longer files are associated with higher accuracy in 5 out of 6 brackets. This outcome is contrary to expectations, as longer files typically have greater complexity and would be expected to degrade performance. Notably, the performance gap reaches nearly 40% in the 512K bracket.

**Topics** Figure 4 shows a radial pie chart illustrating Gemini 1.5 Pro's performance on the six most frequent topics in LongCodeQA. The angle of each slice corresponds to the topic frequency, while the radius indicates the model's accuracy. For example, "Software Development" is the most significant slice (265 samples) with a 70.9% accuracy, whereas "Databases" achieves the highest accuracy (84.2%) but occurs less frequently (38 samples). Overall, accuracy fluctuates around the average of 70%, reflecting the model's relatively stable performance regardless of the repository domain.

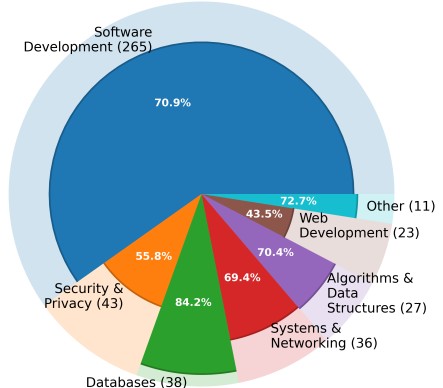

## 5 Conclusion

We introduce LongCodeBench, a comprehensive benchmark that evaluates long-context language models on real-world coding tasks up to one million token windows. By combining code comprehension and repair tasks sourced from GitHub, LCB provides a scalable and realistic framework for assessing model performance on

Figure 4: Distribution of question topics in LongCodeQA, as inferred by an LLM (GPT-4o). The accuracy of Gemini 1.5 Pro for each topic subset is reported and shown as the respective slice radius.

long-context input. Our results show that current LCLMs often degrade performance at scale, revealing a gap between claimed and effective context capabilities. LCB highlights these challenges and offers a foundation for advancing long-context modeling in realistic software engineering settings.

## Acknowledgments

We thank Luca Rigazio and Yoshikuni Sato for their valuable feedback and support during the development of LongCodeBench. We also acknowledge CINECA for providing computational resources and infrastructure support—through access to the Leonardo supercomputer—that made the large-scale evaluations possible. Finally, we thank the anonymous reviewers for their constructive feedback that helped improve the quality of this work.

## Ethics statement

This research uses only publicly available data from open-source GitHub repositories. No personal or sensitive information is included. All experiments comply with the terms of service of the used platforms and APIs. We will release our dataset and code to support transparency and reproducibility.

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

# A  Further analysis on data collection

Here, we report additional experiments that guide the design of LCB's data collection.

## A.1  Performance on the small-scale dataset before and after the first filter

In this section, we analyze the impact of the initial filtering step on a small-scale dataset of around 60 samples. Initially, this dataset is generated by converting GitHub issues into multiple-choice questions using the prompt in Appendix B.1. Then, we assess model accuracy both before and after applying the prompt in Appendix B.2, which filters out questions that do not require detailed, repository-specific knowledge. Results in Table 4 indicate a substantial accuracy drop after filtering, confirming the importance and effectiveness of this step in refining the dataset quality.

| Model | Max. Context Length | Accuracy (%) before filtering | Accuracy (%) after filtering |
|---|---|---|---|
| Jamba 1.5 - 400B Large | 128K | 92.9 | 62.3 |
| Llama 3.1 - 405B Instruct | 256K | 76.2 | 64.6 |

Table 4: Model accuracy comparison on a small dataset to test the effectiveness of the first filter described in Appendix B. After testing the two models on the initial dataset, we filter out questions that do not require repo-specific knowledge.

## A.2  Performance on the complete dataset before final GPT-4o-based filtering

This section presents the results obtained on the complete **LongCodeQA** dataset following the application of the prompts described in Appendix B.1 and B.2. Before the second filter, the dataset includes questions explicitly identified as requiring repository-specific comprehension, but it still includes questions that might be correctly answered without repository context. Table 5 presents the accuracy of LCLMs on the LongCodeQA task, evaluated at different context lengths ranging from 32K to 1M tokens after filtering. The comparison includes both closed-source models (GPT-4o, Claude 3.5 Sonnet, Gemini 1.5 Pro, and Gemini 2 Flash) and open-source models (Qwen2.5, Jamba 1.5, and Llama 3.1). The results indicate that while most models maintain strong performance at shorter and moderate context lengths, accuracy reduces significantly at extreme lengths, such as the 1M token bracket (e.g., Qwen2.5 dropping to 54.5%). Models like Llama 3.1, Jamba 1.5, and the Gemini series demonstrate stable performance across mid-to-high context lengths, highlighting their robustness.

The last filtering step described in section 3.2 provides the final version presented in the main paper. It exclusively contains challenging questions that only require detailed repository analysis, which we show in Table 3.

| Model | Accuracy (%) after filtering | | | | | |
|---|---|---|---|---|---|---|
| | 32K | 64K | 128K | 256K | 512K | 1M |
| Qwen2.5 - 14B Instruct | 75.8 | 75.3 | 82.5 | 79.4 | 82.0 | 54.5 |
| Jamba 1.5 - 400B Large | 83.2 | 77.6 | 81.1 | 72.9 | 74.2 | - |
| Llama 3.1 - 405B Instruct | 83.2 | 83.0 | 83.6 | - | - | - |
| GPT-4o | 80.7 | 83.5 | 85.0 | - | - | - |
| Gemini 2 Flash | 75.2 | 75.9 | 79.7 | 80.0 | 84.4 | 81.4 |
| Gemini 1.5 Pro | 78.0 | 75.9 | 85.0 | 80.0 | 86.0 | 83.4 |
| Claude 3.5 Sonnet | 76.5 | 77.6 | 84.3 | 80.7 | - | - |

Table 5: Model performance for LongCodeQA task on the complete dataset after applying prompts in Appendix B.1 and B.2.

# B   Prompts

Here are reported the prompts used for generating the **LongCodeQA** dataset and for prompting the LLMs.

## B.1   Prompt to identify issues convertible into questions

This prompt assesses whether a GitHub issue can be reformulated as a multiple-choice question.

```
You are an expert software engineer and teacher. You
are given a closed GitHub issue from a public repository and the comments on
the issue. Read the issue and assess whether it is simple enough that its
resolution boils down to clarifying a concept about the codebase. If so,
reexpress it as a multiple choice question with a correct answer and three
wrong answers.
Issue whose resolution required a pull request or a code change are examples
of issues that are not simple enough for this task.
Moreover, the question should be about both the underlying concept and the
codebase, not a general question about programming, software engineering, or
the language being used.
Examples of issues that are simple enough to be expressed as multiple choice
questions:
- Clarifying the behavior of a function, method, or module.
- Understanding the purpose of a variable or class.
- Explaining the flow of control in a code snippet.
- Explaining why a certain error occurs in a code snippet due to a subtle, but
  wanted, behavior of the library being used.
Examples of issues that are not simple enough to be expressed as multiple
choice questions:
- Fixing a bug in the code.
- Implementing a new feature.
- Suggestions on what to pair with a certain library.
- Installation issues.
- Questions about the project's roadmap.
- Observations relating to software engineering practices.
Please provide reasoning to justify your decision. Be conservative, and only
consider issues as simple enough if it is clear beyond doubt that they are.

Issue {issue["title"]}: {issue["body"]}

Comments: {issue["comments"]}
```

## B.2   Prompt to filter questions requiring repository-specific knowledge

This prompt filters the initially generated questions, including only those that require detailed repository-specific knowledge to be accurately answered.

```
You are provided with a question about the repository {repo}. Your task is not to
answer the question directly but to evaluate whether an accurate answer requires a
detailed, up-to-date understanding of the repository, or if it can be answered
accurately using only your pre-existing knowledge (which may include exposure to the
repository during training) and general programming knowledge.

Please explain your reasoning in detail. At the end of your response, on a new line,
output only 'Yes' if you believe the question requires repository-specific knowledge
to answer accurately, or 'No' if you believe it can be answered correctly without
directly consulting the repository.
```

```
Here is the question:
{question and answers}
```

### B.3 Prompt for LongCodeQA task

This prompt is used to answer multiple choice questions based on repository analysis.

```
You are going to be provided the content of a repository and a question about it.
Provide the answer to the question by stating ONLY the letter associated to the
question.

Repository:
{repo_text}

Question:
{question}
```

### B.4 Prompt for LongSWE-Bench task

This prompt generates a patch file to address a coding issue described in a given repository.

```
You will be provided with a partial code base and an issue statement explaining a
problem to resolve.
<issue>
    {issue_body}
</issue>

    {repo_body}

Here is an example of a patch file. It consists of changes to the code base. It
specifies the file names, the line numbers of each change, and the removed and added
lines. A single patch file can contain changes to multiple files.

<patch>
--- a/file.py
+++ b/file.py
@@ -1,27 +1,35 @@
def euclidean(a, b):
-    while b:
-        a, b = b, a % b
-    return a
+    if b == 0:
+        return a
+    return euclidean(b, a % b)

[...truncated for brevity...]
</patch>

I need you to solve the provided issue by generating a single patch file that I can
apply directly to this repository using git apply. Please respond with a single patch
file in the format shown above.

Respond below:
```

## C   Inference time and cost

Table 6 details the inference cost and resource requirements for the models evaluated on LongCodeBench. For self-hosted models, we report both the GPU configuration and the total runtime. For example, Qwen2.5 was run on 8×A100 80GB GPUs for LongSWE-Bench, completing inference in 12 hours, and on 4×A100 64GB GPUs for LongCodeQA, taking 35 hours. In contrast, API-based models show varied costs depending on the provider and context length. Claude 3.5 Sonnet incurs the highest cost, totaling 100 USD for LongCodeQA and 140 USD for LongSWE-Bench. GPT-4o shows lower costs at 50 USD per task. Overall, these figures underscore the high computational and financial burden of evaluating LCLMs on long-context tasks, especially as context windows approach one million tokens.

|  |  | Resources / Provider | LongCodeQA | LongSWE-Bench |
|---|---|---|---|---|
| Self-hosted | Qwen2.5 - 14B Instruct | 8 NVIDIA A100 80GB | - | 12H |
|  |  | 4 NVIDIA A100 64GB | 35H | - |
|  | Jamba 1.5 Large | 20 NVIDIA A100 64GB | 40H | - |
|  | Llama 3.1 - 405B Instruct | 20 NVIDIA A100 64GB | 40H | - |
| Hosted | Jamba 1.5 Large | Amazon Bedrock | - | 70USD |
|  | Llama 3.1 - 405B Instruct | Amazon Bedrock | - | 30USD |
|  | GPT-4o | OpenAI | 50USD | 50USD |
|  | Gemini 2 Flash | Google AI | 20USD | 30USD |
|  | Gemini 1.5 Pro | Google AI | 300USD | 700USD |
|  | Claude 3.5 Sonnet | Anthropic | 100USD | 140USD |

Table 6: Inference cost and time on LCB for the models used in the experiments. For self-hosted models, GPU configuration and total inference time are reported; hosted (API-based) models include provider and total inference cost in USD.

## D   Comparison with SWE-Bench

The LongSWE-Bench task is inspired by Jimenez et al. (2024) but extends it to a significantly longer context scenario. Specifically, we increase the maximum context length to one million tokens and introduce a scalable, granular evaluation across multiple length brackets.

Another key difference with SWE-Bench is in the repository file selection for the context. SWE-Bench uses the BM25 (Robertson & Zaragoza, 2009) retrieval algorithm, we always include the ground-truth files and then add random files as a distraction. As observed by the original SWE-Bench authors, the inclusion or omission of ground-truth files in the prompt creates a sizable gap in performance. This implies that the benchmark results are a combined measure of the models' performance and the accuracy of the retrieval algorithm. Our decision to always provide ground-truth files as context is for the objective of ensuring the benchmark is an unbiased estimator of exclusively models' performance.

For files provided as additional context, we rely on random retrieval for statistical reasons. The creation of samples through relevance-based retrieval of documents introduces bias in the data, as large codebases or other forms of long documents are rarely provided after filtering for non-relevant documents. Random selection, instead, is a form of uniform sampling. This way, we reduce bias at the cost of a higher variance. However, variance can be controlled by providing a sufficiently large number of instances. In this context, random retrieval provides the additional benefit of allowing us to retrieve more samples per issue, as a different selection of the majority of files in a prompt changes the problem in meaningful ways. Figures 5 and 6 show the difference in variance by selecting files randomly or with the BM25 retrieval algorithm. The plots are supported by an analysis of relation between context length and resolution rate through logistic regression. The random retrieval case exhibits a $p$-value of 0.005, which is statistically significant. On the contrary, the $p$-value for BM25 is larger than 0.01.

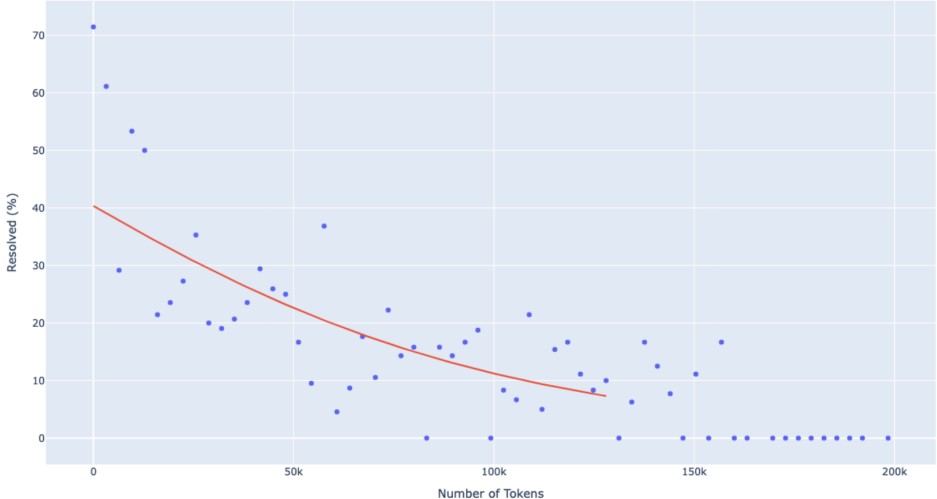

Figure 5: Percentage of solved issues by Claude 3.5 Sonnet on a smaller version of LongSWE-Bench, grouped by bins of equal length range. The red line indicates a trend extracted through logistic regression, with a $p$-value of 0.005—below the threshold (0.01) of statistical significance.

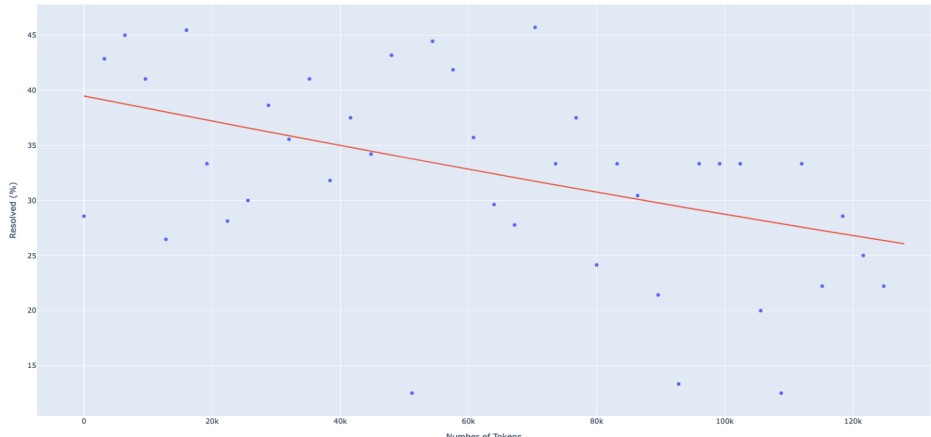

Figure 6: Percentage of solved issues by Claude 3.5 Sonnet on a smaller version of LongSWE-Bench in which non-oracle files are retrieved with BM25, grouped by bins of equal length range. The red line indicates a trend extracted through logistic regression with a $p$-value of 0.025—above the threshold (0.01) of statistical significance.

