# OpenReview forum: "LongCodeBench: Evaluating Coding LLMs at 1M Context Windows"
_colmweb.org/COLM/2025/Conference — COLM 2025_

### Official Review · Reviewer_YF7D · 2025-05-11

**Rating:** 6
**Confidence:** 4
**Ethics Flag:** 1

**Summary:**

LongCodeBench proposes a dual‑task benchmark (code QA + bug‑fix) drawn from real GitHub issues and scaled up to 1 M‑token inputs, a window far beyond today’s public code benchmarks such as SWE‑Bench (≈50 K), Long Code Arena (256 K) arXiv, and generic long‑context suites like RULER (200 K) or HELMET (128 K). Early results show all seven tested frontier LLMs—open and closed—suffer large accuracy drops as context grows, highlighting a gap between advertised and effective long‑context ability. The work is incremental in concept (larger window, similar tasks) yet fills an unoccupied scale niche. Its main weaknesses are (i) no explicit contamination safeguards unlike LiveCodeBench, and (ii) omission of retrieval‑augmented baselines that power many real coding assistants (e.g., RAP‑Gen, MarsCode Agent). Overall, the benchmark is useful for stress‑testing million‑token claims—particularly relevant as models such as Gemini 1.5 advertise that window —but its evaluation protocol needs tightening to ensure fairness and practical relevance.

**Reasons To Accept:**

1. Closes the scale gap. Existing code suites top out at ~50 K tokens for SWE‑Bench and 256 K for Long Code Arena, while LongCodeBench pushes the same tasks to 1 M tokens—finally matching the 1 M‑token window now advertised for models such as Google Gemini 1.5

2. Combines two realistic task types. By pairing repository‑level QA with bug‑fix generation drawn from real GitHub issues, the benchmark captures both comprehension and code‑editing workflows that synthetic “needle‑in‑a‑haystack” or retrieval‑only tests (e.g., RULER / NIAH) overlook.

3. Stratified context brackets aid diagnosis. Six bins from 32 K to 1 M let researchers see exactly where performance begins to fall off. This is a nuance not provided by single‑length suites

Overall, even if i combined SWE‑Bench for bug‑fixing and Long Code Arena for repository‑level Q&A, this benchmark still adds fine‑grained length ladder and gradient signal for weaker models.

4. Transparent compute accounting. The paper reports GPU setups and API costs for every model evaluated, giving practitioners concrete estimates rather than hidden “black‑box” numbers.

**Reasons To Reject:**

1. Conceptually incremental: The benchmark largely scales the SWE‑Bench bug‑fix format and Long Code Arena’s multi‑file tasks upward, so novelty lies more in size than in fresh methodology.

2. No formal contamination defences. Unlike LiveCodeBench’s rolling time‑split and hash logs or BigCodeBench’s provider attestations and overlap scans, LongCodeBench offers no systematic guard against train‑test leakage, risking inflated scores.

3. Retrieval blind spot:. Many production systems rely on retrieval‑augmented pipelines (e.g. cursor), yet the benchmark evaluates only “feed‑the‑whole‑repo” baselines, so it does not reflect dominant real‑world strategies.

---

> ### Author Response · Authors · 2025-06-02
>
> Thank you for the thoughtful review. While our benchmark builds upon prior formats, its novelty lies in scaling evaluation to million-token contexts and unifying comprehension and generation tasks under long-context constraints. With it, we raise awareness of the challenges and provide a rigorous tool to measure progress in this underexplored setting. We address other key concerns below:
>
> **Data Contamination:** We acknowledge this challenge and note that ensuring contamination-free datasets from real-world GitHub issues is inherently difficult. We address this important concern by providing our complete data collection pipeline, enabling easy replication and allowing us or the community to release updated versions as models evolve. This transparency facilitates ongoing contamination monitoring.
>
> **Retrieval-Augmented Baselines:** We include results for Claude 3.5 Sonnet with RAG to provide insight into retrieval-augmented behavior (scores represent percentages of correct answers/resolved issues). However, we deliberately focus on long-context evaluation rather than RAG optimization. We view these as distinct approaches to handling large documents and our benchmark aims to test pure long-context capabilities rather than establish optimal retrieval strategies. Including RAG would introduce confounding variables around prompt structure and retrieval techniques that would obscure the core long-context evaluation.
>
> | Task | 32K | 64K | 128K | 256K | 512K | 1M |
> | - | - | - | - | - | - | - |
> | **LongSWE-Bench** | 23 | 23 | 22 | 21 | 16 | 22 |
> | **LongCodeQA** | 25.55 | 31.18 | 24.83 | 25.95 | 17.19 | 12.77 |
>
> We believe the results are insightful in this case, especially the notable performance in LongSWE-Bench compared to the severe loss of accuracy in LongCodeQA. However, we cannot regard the scores as representative of general RAG approaches, since they are likely dependent on the retrieval parameters we adopted: a selection of the 10 most relevant files included in the prompt.
>
> We believe our benchmark provides a valuable source of assessment for the many other RAG approaches that are possible and yet to be explored. We will include this discussion in the final version of our work.

---

### Official Review · Reviewer_rfFo · 2025-05-12

**Rating:** 6
**Confidence:** 4
**Ethics Flag:** 1

**Summary:**

This paper introduces LongCodeBench (LCB), a novel benchmark for evaluating long-context language models (LCLMs) on code comprehension and repair tasks, with context lengths up to 1M tokens. By grounding tasks in real-world GitHub issues (via LongCodeQA for QA and LongSWE-Bench for bug fixing), LCB addresses a critical gap in existing benchmarks, which often rely on synthetic or shorter-context evaluations. The authors demonstrate significant performance degradation across models as context scales, highlighting the challenges of long-context modeling. The work contributes a scalable, realistic framework for evaluating LCLMs in software engineering scenarios.

**Reasons To Accept:**

1. LCB fills an important gap by focusing on code-related tasks requiring long-context understanding, a practical yet understudied application of LCLMs. The integration of real GitHub issues ensures task realism.

2. The data collection and validation pipeline  ensures high-quality data.

**Reasons To Reject:**

1. Open-source models (e.g., Qwen2.5, Llama 3.1) perform poorly on LongSWE-Bench (0% success). The paper does not investigate whether this stems from architectural limitations, training data gaps, or insufficient optimization for long contexts. A discussion of these factors would add depth. It is better to expand the experimental analysis to include more code-related LLMs, such as the Qwen2.5-Coder series models.

2. The paper evaluates models in a zero-shot setting but does not explore whether fine-tuning on long-context code tasks could bridge performance gaps, limiting insights into adaptability. It is better to add analysis of fine-tuning experiments to validate whether model performance can be improved through fine-tuning.

---

> ### Author Response · Authors · 2025-06-03
>
> Thank you for your review and feedback. We appreciate your recognition of LCB's contribution to addressing the gap in long-context code evaluation and the quality of our data collection pipeline. We would like to address your concerns regarding the experimental scope and fine-tuning analysis.
>
> We acknowledge that the poor performance of open-source models on LongSWE-Bench warrants deeper investigation. The primary focus of our work is on long-context models rather than general LLMs or code-specific models. This constrains our experimental scope, as there are currently very few fine-tuned models capable of handling the large context windows targeted by our benchmark.
>
> We believe it is interesting to evaluate an open-source model specifically fine-tuned for coding as you suggested. We report results for the Qwen2.5-Coder series models, whose context window as provided by API services is restricted to 32K tokens.
>
> | Task | Score (%) |
> | - | - |
> | **LongSWE-Bench** | 8 |
> | **LongCodeQA** | 60%* |
> (*) The score on LongCodeQA was obtained from a subset of questions, as many of them surpassed the 32K limit during generation of the output response.
>
> The results provide insight into code-specific model performance. We must exercise caution in extrapolating conclusions about the behavior of fine-tuned coding models in true long-context settings, given the substantial context length limitations. Exploring fine-tuning strategies is a natural and important next step, which we plan to investigate in future work. We will clarify this in the revised work.
>
> We remain open to further discussion and welcome additional suggestions for strengthening our evaluation framework.

---

### Official Review · Reviewer_5MXK · 2025-05-16

**Rating:** 6
**Confidence:** 2
**Ethics Flag:** 1

**Summary:**

This work introduces LongCodeBench, which includes evaluations focused on both comprehension and repair capabilities—key aspects for long-context LLMs. The results indicate that even state-of-the-art language models still have room for improvement.

**Reasons To Accept:**

This work is strong in its motivation, engineering efforts, and potential for future applications. It appears to address the limitations of existing long-context benchmarks effectively.

**Reasons To Reject:**

No much novelty in methods. But this is some benchmark work so this should be fine.

---

> ### Author Response · Authors · 2025-06-01
>
> We thank the Reviewer for their time and consideration in evaluating our manuscript.
>
> Our benchmark addresses a critical gap in current assessment methodologies by extending analysis to contexts of unprecedented scale—reaching million-token lengths—while integrating both understanding and generation capabilitie. This work highlights previously underexamined difficulties in long-context processing and establishes a systematic methodology for tracking advancement in this emerging research area.
>
> We are committed to strengthening our work and remain open to any additional feedback regarding aspects of the paper. Should any specific questions or concerns arise upon further consideration, we are happy to address them through revision or discussion.

---

### Official Review · Reviewer_KV24 · 2025-06-03

**Rating:** 7
**Confidence:** 5
**Ethics Flag:** 1

**Summary:**

This paper introduces LongCodeBench (LCB), a benchmark for evaluating long-context language models on coding tasks with context windows up to 1 million tokens. LCB comprises two tasks: LongCodeQA, which tests code comprehension via multiple-choice questions derived from GitHub issues, and LongSWE-Bench, which assesses code repair capability by requiring models to generate patches for real-world bugs. The benchmark uses authentic GitHub data and evaluates models across context lengths from 32K to 1M tokens. The authors demonstrate that model performance degrades with increasing context length (e.g., Claude 3.5 Sonnet’s accuracy on LongSWE-Bench drops from 29% at 32K to 3% at 256K), highlighting challenges in long-context scenarios. The empirical findings provide valuable insights for both model developers and users. Overall, LCB is a significant contribution to evaluating LCLMs in realistic coding scenarios.

**Reasons To Accept:**

* LCB is the first coding benchmark supporting 1M-token contexts, far exceeding prior benchmarks (e.g., SWE-Bench at 50K, HELMET at 128K). This capacity reflects modern software repository complexity, enabling fine-grained analysis across 32K to 1M tokens to disentangle context scaling from model capabilities.
* The paper is well-structured and easy to follow, with rigorous data collection (only 3.99% of issues converted to questions) and clear presentation of results.
* Using authentic GitHub repositories and issues rather than synthetic data enhances the practical relevance of the evaluation results.
* LongCodeQA task provides graded feedback via accuracy metrics, offering a "gradient" for model improvement by allowing weaker LCLMs to achieve non-zero performance.
* The systematic evaluation of seven diverse models across multiple context lengths provides valuable data points for the community. The observed performance drops (e.g., Claude 3.5 Sonnet from 29% to 3%) reveal important limitations of current systems.

**Reasons To Reject:**

* LongSWE-Bench and LongCodeQA are both restricted to Python repositories, which may not generalize to other programming languages (e.g., Java, C++). While Python is widely used, this constraint reduces the benchmark’s applicability to diverse software engineering contexts.
* The paper documents performance degradation patterns but offers limited analysis of underlying mechanisms. It lacks examination of attention patterns, failure modes, or error types that could explain why model performance declines. For example, Figure 3 shows that longer files correlate with better performance, contradicting expectations, but this important finding lacks proper investigation and explanation.
* Some minor writing quality issues (e.g., "challange" typo on line 86)

---

### Decision · Program_Chairs · 2025-07-08

**Decision:**

Accept

**Comment:**

The paper presents LongCodeBench, a new benchmark aimed at evaluating the capacity of large language models to handle long-context scenarios—up to 1 million tokens—in programming tasks, specifically LongCodeQA and LongSWE-Bench. The introduction of realistic, task-driven benchmarks for long-context evaluation is a valuable contribution to the community, as recognized by the reviewers. During the author-reviewers discussion phase, the authors addressed most of the reviewers’ concerns by adding clarifying details and presenting new results, including retrieval-augmented generation (RAG) performance and updated findings for Qwen2.5-Coder. I encourage the authors to include these results in the final manuscript. On this basis, I recommend acceptance.